# Effects of a Long-Term Adapted Judo Program on the Health-Related Physical Fitness of Children with ASD

**DOI:** 10.3390/ijerph192416731

**Published:** 2022-12-13

**Authors:** Emanuela Pierantozzi, José Morales, David H. Fukuda, Vanessa Garcia, Antonia M. Gómez, Myriam Guerra-Balic, Eduardo Carballeira

**Affiliations:** 1Department of Neuroscience, Rehabilitation, Ophthalmology, Genetics, Maternal and Child Health, University of Genoa, 1700 Genoa, Italy; 2Faculty of Psychology, Education Sciences and Sport Blanquerna—Ramon Llull University, 08022 Barcelona, Spain; 3School of Kinesiology & Physical Therapy, University of Central Florida, Orlando, FL 32816, USA; 4Department of Physical Education and Sport, University of A Coruna, 15179 Oleiros, Spain

**Keywords:** combat sports, autism, intellectual disabilities, adapted sports, non-exercise equation, waist circumference

## Abstract

Physical fitness is one of the most important physical and mental health aspects for children with Autism Spectrum Disorder (ASD). This study aimed to test the effects of a long-term adapted judo program on the health-related physical fitness of children with ASD. The participants were recruited from various associations of families and schools for children with special needs. Twenty-one children were assigned to an experimental group and nineteen to a control group. The experimental group participated in a six-month adapted judo program consisting of 90 min of practice each week. Health-related physical fitness was measured using the indicators obtained from the ALPHA-fitness battery, the estimated VO_2_max and the waist/height ratio^0.5^. Changes within and between groups were analyzed using linear mixed models for repeated measures designs and test-retest reliability of tests requiring a maximum score using the Intraclass Correlation Coefficient (ICC). A judo program tailored for children with ASD can improve the cardio-metabolic health and cardiorespiratory fitness of its participants. The problems involved with administering physical aptitude tests that involve maximum effort or performance in children with ASD cast serious doubts on the reproducibility of their results.

## 1. Introduction

Physical activity and exercise can help children and adolescents achieve a suitable degree of physical fitness. Maintaining a certain level of physical fitness is one of the most important factors for children and adolescents’ physical and mental health [1]. The World Health Organization guidelines recommend an average of 60 min/day of moderate-to-vigorous intensity aerobic physical activity and regular muscle-strengthening activity across the week for health benefits in children and adolescents [2,3]. This can be done in various settings, including hobbies, recreation and extracurricular youth sports [4].

Autism spectrum disorder (ASD) is a neurological disorder with an unknown cause that manifests itself in difficulties and deficits associated with communication and social interaction and repetitive and stereotyped behaviors [5]. Children with ASD have been shown to have a lower fitness level than children with typical development [6]. The causal relationship between this difference and ASD is unclear; however, it has been reported that challenges when accessing physical activity programs, bullying, lack of awareness of ASD among service providers or instructors, few adapted program options, or the prioritization of therapeutic interventions limited participation [7,8]. The motivation to practice physical activity regularly in children with ASD is affected by complex physical, cultural and environmental elements [9,10,11]. Subsequently, it has been observed that children with ASD tend to be less active, have more motor deficits and, ultimately, live a more sedentary lifestyle [11]. These factors can lead to many chronic diseases, such as obesity, diabetes and cardiovascular diseases [12].

Furthermore, low levels of physical fitness can also reduce social integration [13]. It has been shown that children with ASD show less interest in play and spontaneous games during leisure time activities than their peers without ASD [14], making it difficult to make friends and, therefore, favoring social isolation. Extracurricular sports activities offer an excellent opportunity to increase children's daily PA, stimulate social interactions and appear to be very helpful in preventing illness and improving fundamental motor skills and development in children with ASD [14]. Indeed, structured leisure time activities have positively affected children and adolescents’ mental and physical health [15].

The physical and mental health benefits of physical activity for individuals with ASD have been extensively documented in the scientific literature in [9,10,11,13,16,17]. Some systematic reviews have linked PA to social and communication improvements [18,19], as well as to improved motor skills [20,21]. Other studies have shown how PA and/or exercise programs can improve physical health [22,23].

Martial arts and combat sports involve activities of moderate to vigorous intensity and feature additional cognitive and emotional components such as concentration and self-control [24]. These types of activities are attractive to young people with ASD because of the repetitive structure of the exercises involved [25]. Research supports the effectiveness of these sports not only in improving motor skills [26,27] but in addressing social behavior. For example, the training of karate katas has been found to significantly improve stereotypical behaviors and social interaction [28,29].

The systematic review by Pečnikar et al. [30] highlights the improvements in health parameters and social skills of people with intellectual disabilities when they participate in adapted judo programs. Judo has led to positive results in short-term programs, including improvements in repetitive behaviors, interaction and social communication, and emotional response [16]. Accordingly, a recent study reported [31] a reduction in aggressive behavior in children with ASD who participated in an adapted judo program after an eight-week intervention [31]. Other research demonstrates the viability and effectiveness of this type of program, which can produce a great deal of acceptance and high rates of enjoyment, with participants often expressing a strong desire to continue to take part in the sport after the program is over [32]. It has been reported that adherence to adapted judo programs is associated with an increase in the volume of moderate to vigorous physical activity [24], a fundamental condition for improving the physical fitness of children and adolescents.

Researchers seeking to measure the physical fitness of individuals with ASD face considerable limitations. These issues include communication difficulties, sensory deficits, poor limb function, delayed growth and motor development, defiant behavior and a lack of understanding or motivation to make the required level of effort or to strive for the best possible performance. All of these factors can influence the reliability of a test [33]. In [34,35], the former tests are more cost-efficient and can be carried out in a familiar environment that favors better performance by the participants, even though the latter are usually more precise. A major obstacle researchers face when measuring physical fitness in individuals with ASD is that there is no evidence that certain tests are feasible or reliable with some population subgroups, such as younger children and those with moderate to severe levels of intellectual disability [33].

Therefore, assessing the health-related physical fitness in children with ASD is a challenge. To the best of our knowledge, there are no validated physical fitness batteries of tests exclusively for children with ASD. Generally, the physical fitness in children with ASD has been measured using adaptations of tests for the general population or via existing batteries of tests such as EUROFIT [36,37]. Other batteries, such as the ALPHA health-related fitness test battery [38], have been validated for the population with Down Syndrome [39] and also have adaptations for preschoolers [40]. The use of such tests is a potential methodology in studies of the field of children with ASD.

Anthropometric measurements have been characterized as a good resource for monitoring cardio-metabolic health in this population because of their ease of administration and the consistency of the measurements they yield. Specifically, BMI has historically been used despite its limitations [41]; however, when measuring on the individual level, there are anthropometric alternatives much more valid than BMI. These alternative measurements are based on the waist-to-height ratio (WHTR) [42] or the allometric index (the waist circumference divided by half the height) (WHT.5R = WC/height0.5). The latter is a measurement of waist circumference independent of height [43]. Meanwhile, the variable of maximal oxygen consumption (VO_2_max) is a robust and well-established indicator of cardiovascular health [44] and has proven effective at predicting premature mortality, regardless of its cause [45]. However, it is challenging to perform laboratory or field tests that measure or estimate VO_2_max following maximal effort exercise in children with ASD [33]. In situations like this, when the measurement or estimation of VO2max through an exercise protocol is not possible, researchers have suggested the use of “non-exercise estimation models” that make calculations using factors such as age, gender, self-reported level of physical activity, body composition and other parameters [46,47], all variables that can be easily recorded in children with ASD. These models have been validated against laboratory measurements of VO_2_max and have reported estimation errors from 3.11 to 5.70 mL/kg/min and goodness of fit from r = 0.50 to r = 0.86 [45].

The wide-ranging evidence of the benefits of physical activity and exercise on the health of children with ASD, and the previously demonstrated efficacy of adapted judo programs in this type of population, led to the adoption of the main objective of this study, which is to assess the effects of a long-term adapted judo program on the health-related physical fitness of children with ASD. The secondary objective was to verify the feasibility and reliability of the indicators used to measure physical fitness in this population.

## 2. Materials and Methods

### 2.1. Participants

The GPOWER v3.1 software program (Bonn FRG, University of Bonn, Department of Psychology, Düsseldorf, Germany) was used to calculate the a priori sample size necessary to obtain a Power (1-ß) > 0.9, a large effect size = 0.6 and a type I error = 0.05, two groups and two measurements, with the result being a required total sample of 32 subjects. Based on previous experiences in judo programs with children and adolescents with ASD, we estimated a dropout rate of 20% of the participants during the intervention. Therefore, we recruited forty children for the current study.

The participants’ mean age was 11.07 (±1.73) years, height 145.9 (±15.81) cm and weight 47.71 (±16.71) kg. All of them were recruited from various associations of families of children with ASD and schools for children with special needs. All participants had been diagnosed with ASD based on the Diagnostic and Statistical Manual of Mental Disorders—Fifth Edition (DSM-5) criteria. The psychological reports provided by the participants indicated intelligence quotients (IQ) ranging from 60 to 70 (mean of 65.4 ± 3.55). Individuals who had been medically advised against physical exercise, previously participated in judo classes or simultaneously participated in extracurricular sports activities were excluded. All subjects were invited to participate in the study voluntarily, and both the participants and their families were informed verbally and in writing about the characteristics of the program. Subsequently, the parents or legal guardians signed the informed consent document, and the children signed an informed consent form in which the objectives and plan of the program were explained. All the protocols applied in this research, including managing the participants’ personal data, comply with the requirements specified in the Declaration of Helsinki of 1975 and its subsequent revisions. This study was approved by the Research Ethics Committee of the Ramon Llull University under document number CER URL_2019_2020_003, and the trial was registered at Clinicaltrials.gov (NCT04523805).

### 2.2. Procedure

This research used a prospective design. The convenience sample was divided into two groups based on their availability and commitment to participate in an adapted judo program over a school year. The experimental group (n = 21: age = 11.1 years ± 1.9; height = 147.0 ± 15.7; weight = 47.7 kg ± 12.5) participated in the adapted judo program for six months, and the control group (n = 19: age = 11.0 year ± 1.5; height = 144.5 cm ± 15.9; weight = 47.6 kg ± 10.2) did not participate in any extracurricular sports activities during this period. Each participant's weight and height were measured using a digital balance (Seca 707, Hamburg, Germany) and a wall-mounted stadiometer (Seca 220, Hamburg, Germany) following standard procedures (stand with heels, buttocks and upper back against stadiometer), with each participant being assessed twice, once at the beginning of the program and again at the end. Body mass and height were used to calculate the body mass index, according to Quetelet (kg/m^2^). All measurements were done under stable conditions and in the same room where the judo sessions were held in Barcelona (Spain) during January 2022 and June 2022.

In order to assess the physical condition of the participants at the two different times of measurement, the ALPHA-fitness battery was administered. This battery of tests evaluates the main components of health-related physical conditioning, including cardiorespiratory fitness, musculoskeletal fitness, body composition and motor fitness. The battery of tests has been shown to have a high degree of versatility in its application to special populations [39]. The reliability of the results of physical fitness tests among the population of children with ASD is subject to a high degree of uncertainty, especially when the sample of participants has a wide range of IQs that includes both midrange and low scores [33]. Therefore, additional steps were taken to ensure the reliability of the tests that involve motivation or require maximum effort. In order to confirm the consistency of the responses, intra-session repetitions of the muscle strength tests and the cardiorespiratory fitness test were carried out 48 h apart. In addition, VO_2_max was estimated through a non-exercise equation (NEXE) [46] that uses age, basal heart rate, waist circumference, heart rate and time and intensity of weekly physical activity to estimate cardiovascular capacity. Differences in the factors used to calculate the NEXE have been shown to explain a substantial proportion of the variance in maximal oxygen consumption values among populations of different ages [48,49,50].

### 2.3. Intervention

The experimental group participated in an adapted judo program over six months. The judo sessions were held in an ample, well-ventilated space suitable for athletic activity in general and judo in particular, so the safety of the participants was guaranteed. The judo equipment required for this project included a tatami mat with a surface area of 120 m^2^, made out of high-density covered foam that helps prevent injuries and ensures that a wide range of activities can be carried out safely. Each participant was outfitted with a judogi (the traditional uniform consisting of a cotton jacket, trousers and a belt).

The 90-min sessions were held once a week. Two judo teachers with 7th and 6th DAN levels led the sessions. One has a degree in Pedagogy, and the other in Sports Science. Furthermore, at least four volunteer judo instructors were available at each session to support. The sessions were divided into three parts following the physiological principles of exercise: warm-up, main exercise and cool-down. The main content of the sessions is included in Table 1.

The instructional methodology applied the principle of gradual progression, beginning with practice to consolidate the concepts learned in the initial lessons before moving on to more complex activities and material. Each participant was allowed to progress at his or her own pace. Learning was based on imitation and guided execution of judo-specific skill patterns [51].

The chosen learning method was imitation, where the instructors exposed the techniques and guided the practice. Very marked routines were based on brief and clear instructions, speaking calmly and with a firm voice. The instructions were objective and refrained from using figurative language or irony. Spontaneous and unexpected behavior changes were monitored and redirected by the judo instructors. They were aware that each participant needed their own time. Instructions were given repetitively and used a wide spectrum of senses, not just verbal signals. The isolated use of sensory instructions, one at a time, can aid perception. For example, the instructor can demonstrate physically with verbal instructions and one time without speaking. At the beginning of the program, pictograms were used, but they were stopped because it was considered that it was not necessary.

#### 2.3.1. ALPHA-Fitness Battery

The ALPHA-fitness battery [38] administered in this study is a well-known instrument for measuring health-related physical fitness. This battery consists of various field tests and is suitable for use with children and adolescents. This study used the high-priority variant of the test, which omits skin-fold measurement (triceps and subscapularis). The version used here included the following tests: (1) the 20 m Shuttle Run Test to assess cardiorespiratory fitness; (2) the handgrip strength test; (3) the standing long jump tests to assess musculoskeletal fitness; (4) BMI; and (5) waist circumference. Verbal instructions and demonstrations were given to the participants before each test element until they understood the tasks. The instructors always attempted to motivate all participants [52]. All the tests were carried out in the space that hosted the judo sessions, guaranteeing the necessary space and safety conditions.

The participants’ aerobic capacity was measured using their 20 m Shuttle Run Test scores. Because some of the children in this study had difficulties understanding the instructions, they were sometimes accompanied during their run by the test observers. The test required children to run between two cones placed 20 m apart at a pace set by sound signals. The test ended when the child could not reach the next cone or gave up because of fatigue. The final distance the children ran was recorded. Musculoskeletal fitness was measured via the muscle strength in their hands and legs. Hand strength was measured by asking participants to apply maximal pressure with their dominant hand on the handgrip dynamometer (TKK 5101; Takey, Tokyo, Japan). The device was adjusted to fit the size of each participant’s hand. The participants performed the test standing up with their dominant arm extended, and the best score of two attempts was recorded. The standing long jump test was used to assess the explosive strength of the lower limbs. The children were asked to jump as far as possible on two feet from a standing position. The score for this test was the distance jumped in centimeters on the best jump of two attempts. BMI was calculated as weight (kg) divided by the height squared (m^2^). Participants’ weights and heights were measured using a standard protocol.

#### 2.3.2. A Non-Exercise Equation to Estimate CRF

In order to estimate VO_2_max (eVO_2_max), we recorded data on each participant’s age, waist circumference (WC) and baseline heart rate (HR_baseline_). In addition, the legal guardians of the participants were asked about the frequency and intensity of their children’s physical activity and the time spent on it. These data were then analyzed using the modified version [46] of the physical activity index (PAI), which was previously published by Kurtze et al. [53]. The first question on the questionnaire was, “How many times do you do physical activity each week?” Respondents choose one of the following answers: 0 = never or less than one day a week; 1 = at least one day a week; 2 = two to three days a week; 3 = almost every day. The second question was, “How much time do you spend on each session?” Respondents choose one of the following answers: 1 = less than 30 min; 1.5 = more than 30 min. The third question was, “What is the intensity of the physical activity you do?” Respondents can choose among the following answers: 0 = very soft or soft; 5 = heavy breathing and sweating; 10 = maximum intensity, near exhaustion.

In order to gather data on the HR_baseline_, a finger pulse oximeter (Lifesense LS1-9R, Nonin Medical Inc., Plymouth, MN, United States) was provided to the family of each participant. Family members were instructed to measure for a minute immediately after waking up in the morning for at least four days. The average of the four-day HR_baseline_ results was entered into the formula before and after the intervention.

The participants’ waist circumference (WC) was evaluated in accordance with previously published instructions [54]. Briefly, WC was measured with participants standing with their feet shoulder-width apart using a standard non-elastic anthropometric tape measure. WC measurements were taken to the nearest 0.1 cm, midway between the lower rib and the iliac crest, near the level of the umbilicus, after a gentle exhalation. A trained researcher made two measurements and calculated the mean unless the two values differed by more than 0.5 cm, in which case one more measurement was taken. Waist circumference and WHT.5R were used for analysis.

### 2.4. Statistical Analysis

Descriptive testing data are shown as mean ± standard deviation. The subtests of the ALPHA-fitness battery that require a maximum effort or maximum performance (20 m Shuttle Run Test, handgrip strength, and the standing broad jump) were each recorded twice in a single testing session (each time the best score of two attempts was recorded), except for the 20 m Shuttle Run Test, which was repeated after 48 h. In order to verify test-retest reliability, the Intraclass Correlation Coefficient ICC (bidirectional mixed model, absolute agreement, single measures) was used. ICC values of less than 0.5, from 0.5 to 0.75, from 0.75 to 0.9, and greater than 0.90 indicate poor, moderate, good, and excellent reliability, respectively [55]. ICC calculations were carried out using the Statistical Package for Social Science, version 24.0 (SPSS, Inc., Chicago, IL, USA).

Changes within and between groups were analyzed using linear mixed models for repeated measure designs when all assumptions were met. Normality of the residuals was analyzed using the Shapiro–Wilk test for every variable and revealed no deviations from a normal distribution for WC and WHT.5R. However, HR_baseline_ and eVO_2_max residuals were not normal. Homoscedasticity was checked by plotting the residuals-predicted value [56], and we found that residuals were constant across the predicted values for most of the derived variables but not for HR_baseline_ or eVO_2_max. Therefore, we employed a nonparametric ANOVA-type statistical test for the latter variables. The alpha level was set at *p* < 0.05 for all the analyses.

We employed the module GAMLj for the linear mixed model analyses. This module uses the R formulation of random effects, as implemented by the lme4 R package in jamovi software [57]. GAMLj estimates variance components with restricted (residual) maximum likelihood, which, unlike earlier maximum likelihood estimation, produces unbiased estimates of variance and covariance parameters. The inter-subject group factor (EXP or CON), the intrasubject time factor (PRE and POST tests), and the interaction (GROUP × TIME) were set as fixed effects, and participants’ intercepts were set as a random effect. Changes were evaluated using the β coefficients and a corresponding 95% confidence interval (CI), representing a non-standardized effect size. Between-group changes were assessed using the estimated parameter with a 95% CI of the interaction between the fixed effect of the model. When a significant interaction was detected, we carried out a simple effects analysis of the within-group effect of the time of measurement. The standardized mean difference between PRE and POST was calculated as the mean change score divided by the SD of the change score, termed Cohen’s dz [58], and it was corrected by Hedges’ g to account for the small sample sizes. Cohen’s dz effect was presented with a 90% CI and qualitatively interpreted as trivial if dz < 0.20, small if 0.20 ≤ dz < 0.50, medium if 0.50 ≤ dz < 0.80, large if 0.80 ≤ dz < 1.30 and very large if dz ≥ 1.30 [59].

The effect of the intervention on HR_baseline_ and eVO_2_max was analyzed using nparLD (nonparametric analysis of longitudinal data in factorial experiments) with the R software package [60]. This package calculates nonparametric ANOVA-type statistics (group × time) and uses ranks to calculate relative marginal effects. It was chosen because, unlike traditional nonparametric tests, this test provides information on the effect of each factor and the interaction between them. When a significant interaction was detected, we carried out a simple effects analysis of the within-group effect of the time of measurement. Within-subject changes were analyzed through stochastic superiority (A_post-pre_), which represents the probability that a randomly selected score from the post-intervention will be greater than a randomly selected score from the pre-intervention. Probability values equal to or higher than 0.56, 0.64 and 0.71 when approaching one or values equal to or lower than 0.44, 0.36 and 0.29 when approaching 0 for A_post-pre_ were regarded as small, medium and large values, respectively [61].

## 3. Results

Table 2 shows the descriptive statistics for the ALPHA-fitness battery of tests and the ICC, corresponding to the ICC test-retest values. All ICC scores on the tests requiring maximum effort were poor (<0.5): 20 m Shuttle Run Test (ICC = 0.21), handgrip strength (0.16) and the standing broad jump (0.48). Therefore, it was decided not to use the results of these tests because inconsistency in the results gathered with the sample would make it difficult to interpret any possible changes during data analysis.

There was no difference before the intervention between the study groups in age, height, weight, WC, WHT.5R, HR_baseline,_ and eVO_2_max. Table 3 presents the effects of the fixed factors obtained after analyzing the WC and the WHT.5R using the mixed linear model. Effects of time of measurement and time × group interaction were found for both variables. In the analysis of the simple effects, we found that the EXP group reduced their WC (PRE: 72.5 ± 7.5 cm vs. POST: 70.6 ± 6.9 cm; coefficient: −1.86 cm, 95%CI [−2.63; −1.09]; t38 = −4.90 , *p* < 0.001; dz = −0.98, 90% CI [−1.23; −0.81], large), whereas there were no changes in the CON group (PRE: 73.5 ± 4.9 cm vs. POST: 73.8 ± 5.1 cm; coefficient: 0.32 cm, 95%CI [−0.49, 1.23], t38 = 0.79, *p* = 0.433, dz = 0.19, 90% CI [0.07, 0.33], negligible). However, simple effects analysis showed a reduction in WHT.5R in both the EXP group (PRE: 0.60 ± 0.05 vs. POST: 0.58 ± 0.05; coefficient: −0.024, 95%CI [−0.030; −0.019]; t38 = −7.83, *p* < 0.001; dz = −1.64, 90% CI [−2.04; −1.38], very large) and the CON group (PRE: 0.61 ± 0.04 vs. POST: 0.61 ± 0.04; coefficient: −0.008, 95%CI [−0.015, −0.002], t38 = −2.51, *p* = 0.016, dz = −0.58, 90% CI [−0.79, −0.42], medium).

Figure 1 shows the effects of the fixed factors on eVO_2_max and HR_baseline_ after nonparametric analysis with ANOVA-type statistical analysis. There was an effect of the time of measurement on both variables, and there was a time × group interaction. Simple effects analysis showed an increase in eVO2max in the EXP group (PRE: 53.5 ± 7.4 mL/kg/min vs. POST: 55.2 ± 7.5 mL/kg/min; F_1,∞_ = 73.96, *p* < 0.001; A_post-pre_ = 66.6%, medium), whereas there were no changes in the CON group (PRE: 54.2 ± 6.2 mL/kg/min vs. POST: 54.2 ± 6.2 mL/kg/min; F_1,∞_ = 0.26, *p* = 0.609; A_post-pre_ = 51.4%, negligible). However, there were reductions in HR_baseline_ in both EXP (PRE: 73.60 ± 7.0 ppm vs. POST: 68.3 ± 4.4 ppm; F_1,∞_ = 32.50, *p* < 0.001) and CON (PRE: 72.3 ± 7.8 ppm vs. POST: 70.6 ± 5.5; F_1,∞_ = 32.50, *p* = 0.018), although the effect size was greater in EXP (A_post-pre_ = 28.0%, medium) than CON (A_post-pre_ = 45.6%, negligible).

## 4. Discussion

This study investigated the effects of an adapted judo program on the health-related physical fitness of a group of children with ASD. Participation in the program was linked to greater improvements in body composition parameters (i.e., waist and WHT.5R) and cardiorespiratory fitness (i.e., eVO_2_max) in the experimental group than in the control group. The data on muscle strength, evaluated using the Alpha-fitness battery in the pretest, showed a very low degree of reliability, so it was decided not to carry out a post-test due to the difficulty that the interpretation of the data would entail.

Previous studies have discussed how participation in sports programs can improve the physical fitness of children with ASD. For example, a meta-analysis by Healy et al. (2018) [22] highlights that experimental groups tended to outperform control groups, with a large effect (d = 0.81) in muscular strength and endurance outcomes, although they underscored that best results require longer interventions. The meta-analysis and systematic review by Sam et al. [23] found that children and adolescents with ASD tend to show improvements in physical fitness, exercise mastery and social competence after participating in exercise programs. There is a considerable history of studies examining the effects of specific exercise or sports programs on the health-related fitness of children with ASD. In one study, aquatic exercise programs were effective at improving physical conditioning [62,63]. More recent studies in this regard have detailed the effects of participation in a high-intensity exercise program to improve physical fitness [64], and the participation in both structured physical activities [65] and individualized fitness programs [66]. A five-month rhythmic gymnastics intervention for children with intellectual disabilities [67] obtained similar results to those of the present study in improving cardiorespiratory fitness. Another long-term program in which a treadmill exercise routine was carried out over nine months [68] yielded an improvement in body composition similar to our results. However, these improvements do not appear to be duration-dependent as long as a minimum duration of eight weeks is reached. Shorter interventions using different sports have also arrived at results consistent with our study [62,63,63]. For instance, mini-basketball has also been used effectively to improve the physical fitness of preschool children with autism.

Low levels of cardiorespiratory fitness (CRF) have been identified as a potential risk factor for cardiovascular disease (CVD) and all-cause mortality. Conversely, substantial health benefits can be gained through improved CRF, which can be achieved via physical activity and exercise [45]. In this regard, CRF is viewed as an important marker of cardiovascular health and has even been recommended as a new vital sign by the American Heart Association [45]. Indeed, there is evidence that early intervention and prevention strategies that target youth CRF might be associated with maintaining positive health parameters in later life [69]. Furthermore, CRF is an important marker of physical and mental health and academic achievement in youth [49]. Therefore, family members, educators and health professionals caring for children with ASD should evaluate and implement strategies to improve CRF, including participation in adapted judo programs.

Meanwhile, experiences with other interventions that did not limit their focus to any specific sport also reinforce the results of this study. Such programs usually feature physical exercise interventions with a very similar structure to that of judo sessions. In one study, a high-intensity exercise intervention with alternating rest intervals [64], in which squats, jumping jacks and bear crawls were performed in stationary circuit mode, was linked to improvements in CRF and trunk and limb strength. Arslan et al. (2020) [65] also carried out a structured exercise intervention that included balancing and strength exercises, walking and jumps, leading to significant improvements in running speed and agility, balance, standing long jump performance, reaction times, grip strength and flexibility.

Elsewhere, an intervention using individualized fitness programs [66] also improved overall strength and BMI for half of the participants, although the mean BMI did not change significantly. Srinivasan et al. [34] suggested that BMI, waist circumference and skin-fold thickness measurements could be used to assess changes in body composition as an indicator of health during and after exercise interventions in children with ASD. These authors also suggested quantitative measures of physical activity that included heart rate monitoring and accelerometry, as well as qualitative measures of physical activity that included diaries, logs, and questionnaires completed by parents and caregivers. In addition, it has been pointed out that field tests are preferable to laboratory tests when it comes to measuring the physical fitness of children with ASD [35]. It is important to note that using BMI as a measure can lead to biases that affect the final result. BMI calculations consider weight and height, but changes in weight, especially at early ages, may be due to changes in body composition that do not necessarily imply fat mass gain. For this reason, in our study, we have used the WHT.5R index, which is independent of height and has been suggested as the best WC-derived index associated with metabolic health indicators [43]. In our study, a group effect was found as a function of the time of measurement (meaning a group × time interaction) on the WC and the WHT.5R. Specifically, in the EXP group, WC saw a reduction with a large effect size, and the results for WHT.5R indicated a very large effect size, whereas in the CON group, there was no change in WC and only a moderate reduction in WHT.5R.

It is well-established that judo improves health-related physical fitness in children and adolescents [70]. Apart from its positive effects on health and physical fitness, it has also been associated with psychological and social improvements [71]. The specific focus of this study is on the health-related physical fitness improvements that can come from participating in judo. Numerous studies detailing long-term interventions in typically developing individuals have reported improvements. Several studies examining the influence of one-year judo programs have pointed to improvements in various physical fitness and motor domains such as hand strength, flexibility, general coordination [72], the quality of body posture, balance and lower limb muscle strength impulse [73]. Other investigations have compared the benefits of judo with those of other sports [74,75]. A more recent but shorter study [76] found significant improvements in CRF and body composition among an experimental group of obese children who had participated in a recreational judo program than in a control group. Physical exercise interventions that replicate situations similar to judo sessions have generally been shown to be effective in improving the health of children with ASD. Such exercise sessions are characterized by alternating high- and low-intensity phases, individualized attention, respect for the pace of each participant, explicit verbal and visual instructions, tactile guidance, continuous repetitions and feedback for reinforcement.

Prior research on the effects of judo on the physical fitness of individuals with ASD is somewhat scarce, and the existing studies have always used a much shorter intervention time. The focuses of these earlier studies have mainly been on analyzing the levels of adherence to the adapted judo program, achieving an increase in the volume of physical activity from moderate to vigorous [24], measuring the levels of acceptance and rates of enjoyment or inspiring a solid desire to continue with the activity [32]. Elsewhere, researchers have sought to gauge the decrease in stress and cortisol levels among young people with ASD who do judo [77]. The present study has addressed evidence of the benefits of adapted sport programs employing a relatively large sample compared to similar studies. The study used a long-term intervention and a research design featuring a control group to improve the integrity of the results. Despite the barriers found to measuring physical condition and delivering judo sessions in children with ASD, the present study showed safe and valid protocols that pretend to encourage education (i.e., teachers and pedagogues) and health professionals (i.e., physiotherapists, occupational therapists and sport coaches) to work in a multidisciplinary and interdisciplinary way to promote physical exercise and its linked benefits to this population.

The most important limitation of this study lies in the impossibility of repeating the strength and aerobic endurance tests in the post-test. The results of <0.5 ICC forced us to rule out performing a post measurement. The ALPHA-fitness battery includes tests that have been used regularly in individuals with ASD, but some authors have warned of issues that can emerge when administering the test, depending on the level of IQ of the participants [33]. If the tests require maximum effort or performance, uncertainties arise. Those responsible for administering the tests are never sure whether the participants have jumped as high as they are capable of, whether they have applied all of their strength in the handgrip test, or whether they have continued the 20 m Shuttle Run Test as long as they were able. The use of VO_2_max has been suggested as an indicator of cardiorespiratory fitness and as a very powerful predictor of general health [45]. The use of a "model without exercise" was considered the best way to estimate the VO_2_max of the sample of this study. This model uses easily accessible measures such as age, gender, resting heart rate, self-reported level of physical activity and body composition [46], all of which are factors that have shown their influence on the mechanisms responsible for VO_2_max. The model is validated and has a very large sample that lends it a great deal of credibility. On the other hand, in the field of body composition, BMI has been discarded as an indicator of cardio-metabolic health, and the WHT.5R index has been used instead [43], as it is considered the most suitable instrument in these cases. These two indicators showed significantly greater improvements in both cardiorespiratory fitness and cardio-metabolic health in the experimental group than in the control group.

## 5. Conclusions

The most important conclusion of this study is that it has shown that an adapted judo program for children with ASD can improve the cardio-metabolic health and cardiorespiratory fitness of its participants. The study also highlights the difficulties involved in applying physical fitness tests that involve maximum effort or performance in individuals with ASD, because of doubts about their reproducibility.

The study provides additional support for the monitoring of health-related physical fitness in individuals with ASD through methods that estimate the results with easily accessible data such as age, sex, anthropometric data, HR and the self-reported level of physical activity. Such methods facilitate data gathering and help overcome the uncertainties generated by the application of physical tests to this type of population.

These health improvements are further evidence in favor of the use of judo programs as a complementary intervention to improve cardiovascular risk level and physical fitness in children with ASD. Further studies should investigate the dose-response relationships of judo training with the aim of reducing cardiovascular risk and improving fitness in this population.

## Figures and Tables

**Figure 1 ijerph-19-16731-f001:**
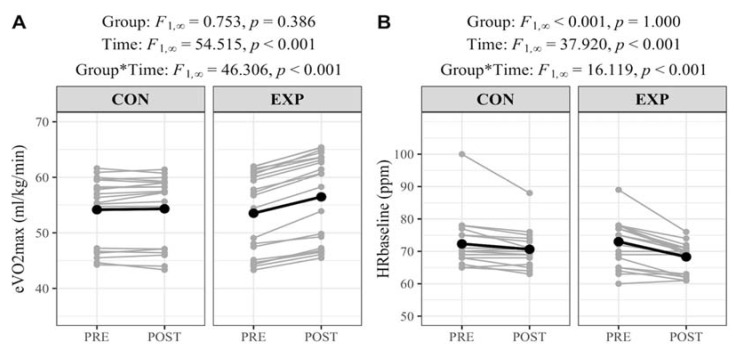
Individual responses and effects of fixed factors on eVO2_max_ and HR_baseline_ before and after the intervention.

**Table 1 ijerph-19-16731-t001:** Contents and temporal distribution of the adapted judo sessions.

Content	Time (min)
Different types of movements and falling techniques (from walking in all directions to change of direction activities, from stable movements to unstable movements).	15–20
Judo analytical techniques and judo games (progressively increasing body contact with games, simplifying movements to focus on essential judo movements).	25–30
Ground control techniques and throws (add technical details incrementally to already known movements, progression from repetitive movements to those more relevant to the understanding and purpose of judo).	25–30
Repetitions of basic movements in different directions and planes (pulling, pushing, holding, lifting).	20–30

**Table 2 ijerph-19-16731-t002:** Descriptive statistics and reliability of the ALPHA-fitness battery of tests.

TEST	Mean (±Standard Deviation)	ICC
20 m Shuttle Run Test	3.45 periods (±1.75)	0.21
Handgrip strength	15.87 kg (±4.95)	0.16
Standing broad jump	127 cm (±0.33)	0.48
BMI	22.23 kg/m^2^ (±2.33)	0.98
Waist circumference	72.95 cm (±6.34)	0.99

**Table 3 ijerph-19-16731-t003:** Fixed effects of the variables WC and WHT.5R.

Variable	Effect	Estimate	Lower 95% CI	Upper 95% CI	t_38_	*p*
Waist Circumference (cm)	(Intercept)	72.59	70.67	74.51	73.94	<0.001
	CON vs. EXP	2.08	−1.76	5.93	1.06	0.295
	POST vs. PRE	−0.77	−0.23	−1.31	−2.80	0.008
	CON vs. EXP POST vs. PRE	2.17	1.10	3.25	3.95	<0.001
WHT.5R (m × m^−1^)	(Intercept)	0.598	0.584	0.611	87.23	<0.001
	CON vs. EXP	0.022	−0.005	0.049	1.62	0.113
	POST vs. PRE	−0.016	−0.020	−0.012	−7.21	<0.001
	CON vs. EXP POST vs. PRE	0.016	0.007	0.025	3.58	<0.001

## Data Availability

Data available at https://doi.org/10.6084/m9.figshare.20465337.

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
