# Peer review of "Effects of a Long-Term Adapted Judo Program on the Health-Related Physical Fitness of Children with ASD"

_ijerph, 2022, doi:10.3390/ijerph192416731_

Round 1
Reviewer 1 Report
Thank you for the authors for submitting this manuscript, the study is interesting and highlights a role for martial arts and physical activity for autistic children. While there are several strengths of the study, there also are a number of areas which require further edits and consideration. Specifically, the introduction would benefit from further information about knowledge of physical activity and the broader forces which impact participation. Specific info is included as part of this feedback.
Introduction:
In addition to the health benefits, the paper would be improved if the authors added more detail and nuance to this discussion, by highlighting other benefits of PA along with the added barriers. Currently, the does not present sufficient background about these issues, despite available literature.
Structure, routine, sense of accomplishment, making friends:
Gregor, S., Bruni, N., Grkinic, P., Schwartz, L., McDonald, A., Thille, P., Gabison, S., Gibson, B. E., & Jachyra, P. (2018). Parents’ perspectives of physical activity participation among Canadian adolescents with Autism Spectrum Disorder. Research in Autism Spectrum Disorders, 48, 53–62. https://doi.org/10.1016/j.rasd.2018.01.007.
The following papers below would help with that discussion and significantly improve the standard of the paper by including them.
Policy Barriers, bullying and exclusion of autistic children from physical activity:
Jachyra, P., Renwick, R., Gladstone, B., Anagnostou, E., & Gibson, B. E. (2021). Physical activity participation among adolescents with autism spectrum disorder. Autism, 25(3), 613–626. https://doi.org/10.1177/1362361320949344
The introduction would be improved further if the authors synthesized it a bit more and explicitly connected the knowledge gaps to the objectives of the study. While some of these elements are there, they are not explicit nor connected and further work in this area would improve the quality of the paper.
The design, methods, and results are well-presented.
Discussion:
Overall the discussion is informative, and well-presented. It is unclear however, how do we further mobilize adapted judo for autistic children given these preliminary findings. Further discussion of this would further improve the quality of the paper. One potential thing to include is to highlight the role of physio therapists or occupational therapists in supporting physical activity and adapted judo for autistic children. Adding this further nuance will strengthen the paper.
Author Response
Editor comments to Author:
Responses to Editor
Please note that:
[E] = comments from Editor
[R1] = comments from Reviewer #1.
[A] = answers from the authors.
{…} = text modified in the revised manuscript.
Reviewer 1:
[R1] = Thank you for the authors for submitting this manuscript, the study is interesting and highlights a role for martial arts and physical activity for autistic children. While there are several strengths of the study, there also are a number of areas which require further edits and consideration. Specifically, the introduction would benefit from further information about knowledge of physical activity and the broader forces which impact participation. Specific info is included as part of this feedback.
[A] = Thank you very much for your suggestions. We really appreciate your effort in reviewing the manuscript.
[R1] = Introduction: In addition to the health benefits, the paper would be improved if the authors added more detail and nuance to this discussion, by highlighting other benefits of PA along with the added barriers. Currently, the does not present sufficient background about these issues, despite available literature.
Structure, routine, sense of accomplishment, making friends:
Gregor, S., Bruni, N., Grkinic, P., Schwartz, L., McDonald, A., Thille, P., Gabison, S., Gibson, B. E., & Jachyra, P. (2018). Parents’ perspectives of physical activity participation among Canadian adolescents with Autism Spectrum Disorder. Research in Autism Spectrum Disorders, 48, 53–62. https://doi.org/10.1016/j.rasd.2018.01.007.
The following papers below would help with that discussion and significantly improve the standard of the paper by including them.
Policy Barriers, bullying and exclusion of autistic children from physical activity:
Jachyra, P., Renwick, R., Gladstone, B., Anagnostou, E., & Gibson, B. E. (2021). Physical activity participation among adolescents with autism spectrum disorder. Autism, 25(3), 613–626. https://doi.org/10.1177/1362361320949344
The introduction would be improved further if the authors synthesized it a bit more and explicitly connected the knowledge gaps to the objectives of the study. While some of these elements are there, they are not explicit nor connected and further work in this area would improve the quality of the paper.
[A] = The authors thank the reviewer for the suggestions that definitely raise the quality of our manuscript. We have added the reference suggested within the text and reordered the ideas to connect the knowledge gaps with the study's objectives. We added this text:
“…however,, it has been reported that challenges when accessing physical activity pro-grams, bullying, lack of awareness of ASD among service providers or instructors, few adapted program options, or the prioritization of therapeutic interventions limited participation [7,8]”
[R1] = The design, methods, and results are well-presented.
[A] = Thank you for your commentaries.
[R1] = Discussion: Overall the discussion is informative, and well-presented. It is unclear however, how do we further mobilize adapted judo for autistic children given these preliminary findings. Further discussion of this would further improve the quality of the paper. One potential thing to include is to highlight the role of physio therapists or occupational therapists in supporting physical activity and adapted judo for autistic children. Adding this further nuance will strengthen the paper.
[A] = Thank you for your suggestion. We have added within the text the importance of promoting adapted sports programs similar to those presented in our paper and encouraged education and health professionals to work together to achieve an effective “physical activation” of children with ASD. Future works with adapted judo programs are warranted to study the social, cognitive, and physical effects of the practice of this sport in adapted programs where children with and without ASD practice together.
“Despite the barriers found to measuring physical condition and delivering judo sessions in children with ASD, the present study showed safe and valid protocols that pretend to encourage education (i.e., teachers and pedagogues) and health professionals (i.e., physiotherapists, occupational therapists, and sport coaches) to work multidisciplinary and interdisciplinary to promote physical exercise and its linked benefits to this population.”
Reviewer 2 Report
With regard to the main question addressed by the research, the work presented seeks to carry out an analysis of the impact of sports practice on children with ASD. To this end, the authors have designed a Judo sports training programme. Given the target profile of the study, in order to carry out the sports sessions it was necessary to adapt the sport of Judo. This has made it possible to measure the effect that Judo can have on children with ASD and to assess the type of health benefit it can have.
With regard to the originality of the study, it should be noted that the authors propose an original study that is also relevant for the scientific community, given that it links two very specific topics such as sport and ASD. Given the theoretical basis of the study, evidence from other studies and authors who have worked along the same lines, and being aware of the difficulty involved in introducing sport to work with this type of profile, the proposed study is a further scientific contribution that helps to reinforce the theory that sport can be used as an alternative to improve different skills in this group.
The paper represents yet another contribution to the scientific community on the impact of sport on people with ASD. In addition, it includes updates related to individual sport for the benefit of health in children with ASD. Judo, being an individual and "minority" sport, represents particular sporting characteristics that help to understand how, from another perspective, different skills can be enhanced to help improve the quality of life of people with ASD.
The chapter on the methodology of the work fits well with the proposal of the study.
There is a good description of the type of sample involved, the procedure, and the tools used to measure the different dimensions. In addition, the statistical analysis on which the research was based is detailed.
The conclusions drawn in the study are coherent with the type of study and with the analysis of the results. The discussion of the work has made it possible to carry out a contrast analysis between the results obtained and the evidence of other authors who have worked along the same lines of study. This has allowed the author to draw more precise conclusions such as those contained in the research work.
The work has been well documented. Scientific literature has been used to justify the proposed research work. The references used are in line with the theoretical basis of the work. The authors have relied on different studies to justify the inclusion of sport in working with people with ASD.
Finally, the figures and tables present the most relevant data from the study, giving greater visibility to those results that have the greatest impact on the study's intervention.
Author Response
Reviewer 2:
[R2] = With regard to the main question addressed by the research, the work presented seeks to carry out an analysis of the impact of sports practice on children with ASD. To this end, the authors have designed a Judo sports training programme. Given the target profile of the study, in order to carry out the sports sessions it was necessary to adapt the sport of Judo. This has made it possible to measure the effect that Judo can have on children with ASD and to assess the type of health benefit it can have.
[A] = Thank you for your commentaries. We really appreciate your effort in reviewing the manuscript.
[R2] = With regard to the originality of the study, it should be noted that the authors propose an original study that is also relevant for the scientific community, given that it links two very specific topics such as sport and ASD. Given the theoretical basis of the study, evidence from other studies and authors who have worked along the same lines, and being aware of the difficulty involved in introducing sport to work with this type of profile, the proposed study is a further scientific contribution that helps to reinforce the theory that sport can be used as an alternative to improve different skills in this group.
The paper represents yet another contribution to the scientific community on the impact of sport on people with ASD. In addition, it includes updates related to individual sport for the benefit of health in children with ASD. Judo, being an individual and "minority" sport, represents particular sporting characteristics that help to understand how, from another perspective, different skills can be enhanced to help improve the quality of life of people with ASD.
[A] = Thank you for your commentaries. We really appreciate your effort in reviewing the manuscript.
[R2] = The chapter on the methodology of the work fits well with the proposal of the study.
[R2] = There is a good description of the type of sample involved, the procedure, and the tools used to measure the different dimensions. In addition, the statistical analysis on which the research was based is detailed.
[R2] = The conclusions drawn in the study are coherent with the type of study and with the analysis of the results. The discussion of the work has made it possible to carry out a contrast analysis between the results obtained and the evidence of other authors who have worked along the same lines of study. This has allowed the author to draw more precise conclusions such as those contained in the research work.
[A] = Thank you for your commentaries. We really appreciate your effort in reviewing the manuscript.
[R2] = The work has been well documented. Scientific literature has been used to justify the proposed research work. The references used are in line with the theoretical basis of the work. The authors have relied on different studies to justify the inclusion of sport in working with people with ASD.
[R2] = Finally, the figures and tables present the most relevant data from the study, giving greater visibility to those results that have the greatest impact on the study's intervention.
[A] = Thank you for your commentaries.
Author Response
Reviewer 3:
[R3] = Thank you for the opportunity to review this paper. The aim of the study was to assess the effects of a long-term adapted judo program on the health-related physical fitness of children with ASD. The secondary objective was to verify the feasibility and reliability of the indicators used to measure physical fitness in this population. It is mostly clear and the most of the conclusions are sound. There is, however, major issues that must be resolved before the study can be accepted for publication.
[A] = Thank you very much. We appreciate your effort in reviewing the manuscript that definitely raise the quality of our manuscript.
[R3] = 1. Introduction
-line 32: provide recommendations on physical activity, World Health Organization, 2020
[A] = The information has been updated to The World Health Organization guidelines of 2020
Bull, F. C., Al-Ansari, S. S., Biddle, S., Borodulin, K., Buman, M. P., Cardon, G., ... & Willumsen, J. F. (2020). World Health Organization 2020 guidelines on physical activity and sedentary behaviour. British journal of sports medicine, 54(24), 1451-1462.
[R3] = -line 42: expand ASD (first use)
[A] = Thank you very much. It has been changed
[R3] = -describe the characteristics of ASD
[A] = Thank you, amended as follows:
Autism spectrum disorder (ASD) is a neurological disorder with an unknown cause that manifests itself in difficulties and deficits associated with communication and social interaction, as well as in repetitive and stereotyped behaviors…
[R3] = -line 47: “(McCoy et al., 2016)” - give a proper reference (number)
[A] = Thank you very much. It has been changed
[R3] = -lines 55-57: “as well as other more complex physical, cultural and environmental elements that make children with ASD more passive and less motivated to do physical activity” - for example, please describe them
[A] = Thank you, amended as follows:
It has been shown that children with ASD show less interest in play and spontaneous games during leisure time activities than their peers without ASD.
We added this reference:
Serrada-Tejeda S, Santos-del-Riego S, May-Benson TA, Pérez-de-Heredia-Torres M. Influence of Ideational Praxis on the Development of Play and Adaptive Behavior of Children with Autism Spectrum Disorder: A Comparative Analysis. Int J Environ Res Public Health. 2021;18(11):5704.
[R3] = -lines 62-75: “Healey et al. [17] found improvements in various areas such as manipulative and locomotor skills, social skills, strength, endurance, and physical fitness in general. The meta-analysis and systematic review by Sam et al. [18] found that children and adolescents with ASD tend to show improvements in physical fitness, exercise mastery and social competence after taking part in exercise programs. There is a considerable history of studies examining the effects of specific exercise or sports programs on the health-related fitness of children with ASD. In one study, an exercise program based on walking on a treadmill significantly improved the body mass index (BMI) of the participants [19], and, elsewhere, aquatic exercise programs have been shown to be effective at improving physical conditioning [20,21]. More recent studies in this regard have detailed the effects of participation in a mini-basketball program with the goal of improving physical fitness and social skills [22], the application of a high intensity exercise program to improve physical fitness [23], and the participation in both structured physical activities [24] and individualized fitness programs [25]” - this should be a part of the discussion section
[A] = Thanks for your suggestion. We have adapted the information in this paragraph and included it in the discussion section.
[R3] = -lines 83-92: „The systematic review by Pečnikar et al. [32] highlights the improvements in health parameters and social skills of people with intellectual disabilities when they participate in adapted judo programs. Judo has led to positive results in short-term programs, including improvements in repetitive behaviors, interaction and social communication, and emotional response [10]. Proof of this are the improvements reported in a study with an eight-week intervention [33] which found a reduction in aggressive behavior in children with ASD who participated in an adapted judo program. Other research demonstrates the viability and effectiveness of this type of program, which can produce a great deal of acceptance and high rates of enjoyment, with participants often expressing a strong desire to continue to take part in the sport after the program is over [34].” - this should be a part of the discussion section
[A] = We believe that this content highlights in the introduction the importance of judo in improving the quality of life of children with ASD. We are aware that it provides detailed information, but since it is the central theme of the article we would like to emphasize this aspect. In the discussion, the same ideas are discussed in depth and contrasted with our results.
[R3] = -lines 101-107: “Srinivasan et al. [36] suggested that methods such as BMI, waist circumference and skinfold thickness measurements could be used to assess changes in body composition, as an indicator of health during and after exercise interventions in children with ASD. These authors also suggested quantitative measures of physical activity that included heart rate monitoring and accelerometry, as well as qualitative measures of physical activity that included diaries, logs, and questionnaires completed by parents and caregivers. In addition, it has been pointed out that field tests are preferable to laboratory tests when it comes to measuring the physical fitness of children with ASD [37].” - this should be a part of the discussion section.
[A] = Thank you for your suggestion. We have moved this paragraph to discussion section.
[R3] = -lines 130-132: “These indices derived from the measurement of waist circumference have been shown to be better at explaining/predicting cardiometabolic risk factors, even when controlling for the effects of age, sex, and ethnicity [45]” irrelevant information
[A] = Thank you for the suggestion. We have removed this paragraph.
[R3] = 2. Procedure. Where were the measurements taken? And what procedures were followed? Please describe your measurements in detail. Please indicate the protocol for taking height and weight data. Also include the brand of each instrument used.
Regarding your variables analysed, you should add reference value in the methodology section (BMI).
[A] = We added this information: "Each participant's weight and height were measured using a digital balance (Seca 707, Hamburg, Germany) and a wall-mounted stadiometer (Seca 220, Hamburg, Germany) following standard procedures (stand with heels, buttocks and upper back against stadiometer) and was assessed twice, once at the beginning of the program and again at the end. Body mass and height were used to calculate the body mass index, according to Quetelet (kg/m2). All measurements were done, under stable conditions and in the same room where the judo sessions were held in Barcelona (Spain) during January 2022 and June 2022.
[R3] = 3. Intervention
Whether methods of AAC were used?
[A] = We have added a paragraph explaining the methods of instruction during the intervention
“The chosen learning method was imitation, where the instructors exposed the techniques and guided the practice. Very marked routines were based on brief and clear instructions, speaking calmly and with a firm voice. The instructions were objective and refrained from using figurative language or irony. Spontaneous and unexpected behavior changes were monitored and redirected by the judo instructors. They were aware that each participant needed their own time. Instructions were given repetitively and used a wide spectrum of senses, not just verbal signals. The isolated use of sensory instructions, one at a time, can aid perception. For example, the instructor can demonstrate physically with verbal instructions and one time without speaking. During the beginning of the program pictograms were used, but they were stopped being used since it was considered that it was not necessary.”
[R3] = 4. ALPHA-fitness battery. Line 237: How was the verification done?
[A] = It was found that the subtests of the ALPHA fitness battery that required maximum effort had very low consistency during the repetitions of the first evaluation. The research team decided to carry out a test retest during the following 48 h to verify the reliability of these measures, the intra-session results were very low, but the inter-session results were worse. Finally, it was decided not to include these results in the post test and to use the estimate of cardiovascular health with other indicators.
[R3] = 5. Statistical Analysis
Line 305: (https://www.jamovi.org/) - give it to references
[A] = Apologies for the missing reference. We have added it to the reference list.
Round 2
Reviewer 1 Report
Great work by the authors to improve the quality of the papers.